# Efficacy of Surgery for the Treatment of Gastric Cancer Liver Metastases: A Systematic Review of the Literature and Meta-Analysis of Prognostic Factors

**DOI:** 10.3390/jcm10051141

**Published:** 2021-03-09

**Authors:** Gianpaolo Marte, Andrea Tufo, Francesca Steccanella, Ester Marra, Piera Federico, Angelica Petrillo, Pietro Maida

**Affiliations:** 1Department of General Surgery, Ospedale del Mare, 80147 Naples, Italy; tufo.andrea@gmail.com (A.T.); fra.steccanella@gmail.com (F.S.); estermarra9@gmail.com (E.M.); p.maida@libero.it (P.M.); 2Medical Oncology Unit, Ospedale del Mare, 80147 Naples, Italy; pierafederico@yahoo.it (P.F.); angelic.petrillo@gmail.com (A.P.)

**Keywords:** gastric cancer, liver metastasis, conversion surgery, hepatectomy, stage iv gastric cancer

## Abstract

Background: In the last 10 years, the management of patients with gastric cancer liver metastases (GCLM) has changed from chemotherapy alone, towards a multidisciplinary treatment with liver surgery playing a leading role. The aim of this systematic review and meta-analysis is to assess the efficacy of hepatectomy for GCLM and to analyze the impact of related prognostic factors on long-term outcomes. Methods: The databases PubMed (Medline), EMBASE, and Google Scholar were searched for relevant articles from January 2010 to September 2020. We included prospective and retrospective studies that reported the outcomes after hepatectomy for GCLM. A systematic review of the literature and meta-analysis of prognostic factors was performed. Results: We included 40 studies, including 1573 participants who underwent hepatic resection for GCLM. Post-operative morbidity and 30-day mortality rates were 24.7% and 1.6%, respectively. One-year, 3-years, and 5-years overall survival (OS) were 72%, 37%, and 26%, respectively. The 1-year, 3-years, and 5-years disease-free survival (DFS) were 44%, 24%, and 22%, respectively. Well-moderately differentiated tumors, pT1–2 and pN0–1 adenocarcinoma, R0 resection, the presence of solitary metastasis, unilobar metastases, metachronous metastasis, and chemotherapy were all strongly positively associated to better OS and DFS. Conclusion: In the present study, we demonstrated that hepatectomy for GCLM is feasible and provides benefits in terms of long-term survival. Identification of patient subgroups that could benefit from surgical treatment is mandatory in a multidisciplinary setting.

## 1. Introduction

Gastric cancer (GC) is the fourth most common cancer worldwide and the second among cancer deaths [1]. Distant metastases are found in 30–35% of patients at their first clinical observation and they spread commonly to the liver (48% of metastatic cancer patients), peritoneum (32%), lung (15%), and bone (12%). Patients with stage IV GC have a median survival of 3 months, which is worst among those with bone and liver metastases (2 months) [2]. According to current guidelines, systemic chemotherapy is recommended as a single modality treatment for stage IV GC [3]. However, despite the development of new molecular targeting agents, the prognosis remains unsatisfactory, with a reported median overall survival (OS) of 13.8 months [4,5].

The role of surgical resection of GC metastases has always been debated. However, in the last 10 years, many studies showed encouraging results. Recently, Yoshida et al. proposed a new classification of stage IV GC, dividing the stage IV in four categories which results in different treatment approaches (Figure 1) [6]. Gastric cancer liver metastases (GCLM) without peritoneal carcinomatosis belong to categories 1 and 2; in those patients, the authors suggest using the so-called “conversion therapy”, which consists of intensive chemotherapy followed by adjuvant surgery if radical resection is achievable. In addition, the last revision of the Japanese Gastric Cancer guidelines stated that hepatectomy can be considered for a subset of patients such as cases with solitary liver metastasis [7]. This approach is also endorsed by the Italian Group on Gastric Cancer Research [8].

However, GC patients often present with metastases in multiple sites and only 0.4–1% of patients have liver metastases amenable to radical resection [7]. Most recent reports in the literature showed promising results after adopting aggressive multidisciplinary management, including surgery, to treat patients with GCLM [9].

In our study, we performed a systematic review of the literature of the last decade to evaluate the OS and disease-free survival (DFS) after hepatectomy for GCLM, and a meta-analysis of the prognostic factors, in order to clarify which patients would benefit more from surgical treatment.

## 2. Materials and Methods

A systematic review protocol was registered at the International Prospective Register of Systematic Reviews (PROSPERO): database registration number CRD42021218350 [10]. This study is reported in compliance with the Transparent Reporting of Systematic Reviews and Meta-analyses (PRISMA) statement [11].

### 2.1. Criteria for Considering Studies for This Review

#### 2.1.1. Type of Studies

We included prospective and retrospective studies reporting survival outcomes after hepatectomy for GCLM. We have selected only studies involving humans and available as full text in English published in the last decade. We excluded case reports, animal and other experimental, as well as purely imaging studies.

#### 2.1.2. Type of Participants

We included studies where all participants who had hepatectomy for GCLM were eligible for upfront radical resection (R0) of both primary tumor and metastasis in the liver. In order to avoid selection bias, studies including participants with metastatic sites other than the liver were excluded.

#### 2.1.3. Type of Interventions and Outcomes

We included only studies in which there were reported short-term outcomes (post-operative morbidity and 30-day mortality) and long-term outcomes (1-, 3-, 5-years OS and DFS) after hepatectomy for GCLM. We investigated the impact of the prognostic factors collected from the studies on OS and DFS.

### 2.2. Search Methods for Identification of Studies

#### 2.2.1. Electronic Searches

Two independent reviewers (G.M. and F.S.) performed a systematic search of the literature, from January 2010 to September 2020. The authors did not consider previous articles as the management of gastric liver metastases has changed significantly over the last decade.

We searched the PubMed (Medline), Cochrane, EMBASE, and Google Scholar databases using MeSH and free text words (tw) for GC and liver metastases.

We performed the search using different combinations of the following keywords: “gastric AND cancer AND hepatectomy”, “gastric AND cancer AND metastases” “gastric AND cancer AND metastasectomy”, “stomach AND cancer AND hepatectomy”, “stomach AND cancer AND metastasectomy”. The same search was then repeated changing the word “cancer” with “carcinoma”, “cancer” with “neoplasm”, “metastases with “metastasis”, and “hepatectomy” with “liver resection”.

#### 2.2.2. Searching Other Resources

We also checked the references of the selected studies in order to find further relevant trials.

### 2.3. Data Collection and Analysis

Two authors (G.M. and F.S.) independently selected studies and two authors (G.M. and A.T.) extracted data from those trials in a pre-piloted data extraction form created using Microsoft Excel (Microsoft, Redmond, WA, USA).

#### Study Selection

Study selection was performed by first screening the titles and abstracts in order to exclude the studies that were clearly not eligible. Then, using the predefined inclusion and exclusion criteria, the full texts of the studies were screened. Figure 2 illustrates the flow-chart diagram of the study selection. 

Then, some papers were excluded after discussion between the two reviewers (G.M. and F.S.), because they were not strictly linked to the topic of the review. 

Ethical approval and informed consent were not needed for this paper as per local rules at our institution.

### 2.4. Literature Search

#### 2.4.1. Data Extraction and Management

Two independent reviewers performed data collection (G.M. and A.T.) and included the following data:Year of publication.Country of recruitment.Study interval (year(s) in which the trial was conducted).Inclusion and exclusion criteria.Population details, such as age, sex, characteristics of the primary tumor, and liver metastases.Outcomes (mentioned in ‘Type of interventions and outcomes’).Details of the prognostic factor(s).

The reviewer extract survival data from tables, directly from the text whenever possible or by manual interpolation in case of data available only in graphs. The clarification of unclear or missing information was done by direct contact with the authors of each study. We solved any differences in opinion through discussion.

#### 2.4.2. Assessment of Risk of Bias in Included Studies 

Due to the nature of this systematic review, the study quality or risk of bias was assessed by the Oxford Centre for Evidence-Based Medicine (CEBM) classification only for descriptive purposes [12].

#### 2.4.3. Data Synthesis

One, three, and five-year OS and DFS were calculated as the proportion of patients alive and free from the tumor at 1, 3, and 5 years and the total of patients included in the study. Median Survival Time (MST) was also calculated. StatsDirect Software (StatsDirect Ltd., Birkenhead, UK) was used to calculate the meta-analysis of proportion [13].

Hazard ratio (HR) with corresponding 95% confidence intervals (95% CIs) were calculated to assess the impact on OS and DFS of the liver metastases related prognostic factors using the inverse variance method with Review Manager 5.4 (Cochrane Collaboration) [14]. 

The HR and its variance were obtained from the study or calculated according to the data presentation: annual mortality rates, survival curves, number of deaths, or percentage freedom from death [15].

A random-effects model was used to perform a meta-analysis due to the clinical heterogeneity among studies. Funnel plots were used to graphically represent publication bias and in order to find asymmetry and any outliers. Heterogeneity across the studies was assessed using the Cochran Q test and/or the I^2^ statistic to measure the degree of variation not attributable to chance alone. This was graded as low (I^2^ < 25%), moderate (I^2^ = 25% to 75%), or high (I^2^ > 75%) [16]. 

A significant *p* value < 0.05 was considered in order to assess statistically significant differences in each analysis. Forest plots showed the results of the current meta-analysis. Calculations were performed by A.T. and verified by G.M.

#### 2.4.4. Subgroup Analysis

We performed a subgroup analysis of the OS and DFS based on ethnicity. One, three, and five-year OS and DFS were calculated as the proportion of patients alive and free from tumor at 1, 3, and 5 years and the total patients included in the study. StatsDirect Software was used to calculate the meta-analysis of proportion [17].

## 3. Results

We identified 2499 references through electronic searches of Medline (n = 1212), EMBASE (n = 551), and Google Scholar (n = 736). We excluded 2439 duplicates and clearly irrelevant references through reading the abstracts. The remaining 60 records were retrieved as full text for further assessment. Then, we discharged the other 4 references (for further details, see the section “Characteristic of excluded studies” below). At least, 40 studies were included in the study and were finally analyzed.

### 3.1. Characteristics of the Included Studies

The 40 included studies [8,18,19,20,21,22,23,24,25,26,27,28,29,30,31,32,33,34,35,36,37,38,39,40,41,42,43,44,45,46,47,48,49,50,51,52,53,54,55,56,57] included 1573 participants who underwent hepatic resection for GCLM. All studies were reported in English. All studies were retrospective analyses. No prospective studies or randomized trials were found (Table 1 and Table 2). The majority of the studies were conducted in Asia (33 studies: 82.5%), whereas only 7 trials included data from Western countries (17.5%).

The median age of the whole population was 64 years old (range 30–89), the majority of whom were men (1050 participants, 66.8%). Two hundred eighty-five participants (30.3%) had pT1–2 and 656 (69.7%) pT3–4 gastric adenocarcinoma in 26 studies reporting this data. Four hundred twenty participants (37%) had pN0–1 and 714 (63%) pN2–3 gastric adenocarcinoma in 25 studies reporting this data. Lymphatic invasion was present in 230 patients (48%) included in 10 studies, whereas 216 patients (53.7%) included in 7 studies had a venous invasion. Nine studies reported the size of the primary tumor: 94 participants (37.3%) had primary tumor <5 cm and 158 (62.3%) >5 cm (median 5.2 cm). The tumor was poorly differentiated in 236 patients (16.4%), whereas it was well or moderately differentiated in 1201 (83.6%) participants in 19 studies reporting this data. Six hundred twelve participants (63.4%) underwent surgical resection for solitary hepatic metastasis and 354 (36.6%) were treated for multiple hepatic metastases in 27 studies reporting this data. Three hundred fifteen participants (78.5%) underwent surgical resection for unilobar hepatic metastases and 86 (21.5%) were treated for bilobar hepatic metastases in 13 studies reporting this data. Eight hundred seventy-six participants (62.3%) underwent surgical resection for synchronous liver metastases and 529 (37.7%) were treated for metachronous hepatic metastases in 33 studies reporting this data.

The median time between gastrectomy and the onset of hepatic metastases was 12.5 months (range 7–135) in 12 studies reporting this data.

The median size of the hepatic metastases was 28 mm (range 17–160 mm) and 52 participants (57.7%) had liver tumor >3 cm and 38 <3cm (42.3%) in 18 studies reporting this data. Seven hundred ninety-nine participants (77%) underwent minor hepatectomy and 239 (23%) major hepatectomy in 18 studies reporting this data. A total of 757 participants (86.3%) underwent R0 hepatic resection, 72 (8.2%) had R1 resection, and 48 (5.5%) R2 in 12 studies reporting this data.

One hundred fifty-one participants (12.3%) underwent neoadjuvant whereas 610 (48.6%) underwent adjuvant chemotherapy in 30 studies reporting this data.

### 3.2. Characteristics of the Excluded Studies, Risk of Bias, and Applicability Concerns

Three studies were excluded because authors included also patients with peritoneal metastasis [58,59,60].

Data from one study were excluded from all the analyses because the authors did not specify if the patients underwent liver resection or other treatments (Radiofrequency ablation (RFA), Microwave ablation, others) [60]. Data from 4 studies were included from the analysis of the short-term outcomes (morbidity and mortality) but excluded from the analysis of the OS, DFS, and the analysis of the prognostic factors because the authors did not specify if the patients underwent liver resection or other treatments [33,39,41,42].

All the studies included in the analysis had a type 2b quality of evidence, according to the Oxford Centre for Evidence-Based Medicine scoring system.

### 3.3. Discrimination Results

#### 3.3.1. Morbidity and Mortality

Surgical resection of GCLM was performed with 24.7% of morbidity from the 19 studies reporting this data (250/1011) and 1.6% of 30-day mortality from the 30 studies reporting this data (22/1338). Short- and long-term outcomes are shown in Table 3.

#### 3.3.2. Survival Data

After a median follow-up of 26 months (range 8–77), the 1-year, 3-year, and 5-year OS were 72% (range 66–77%), 37% (range 31–43%), and 26% (range 21–30%), respectively (Figure 3).

The 1-year, 3-year and 5-year DFS were 44% (range 40–48%), 24% (range 19–29%), and 22% (range 15–31%), respectively (Figure 4).

Subgroup meta-analyses based on ethnicity were performed (Figure 5 and Figure 6). Eastern studies showed better 1-year (75% vs. 59%), (HR 0.38, 0.29–0.49, *p* < 0.00001, I^2^ 78.9%), 3-year (39% vs. 28%) (HR 0.44, 0.32–0.60, *p* < 0.00001, I^2^ 72.3%), and 5 year (27% vs. 19%) (HR 0.36, 0.21–0.60, *p* = 0.0001, I^2^ 66.2%) OS. The results of the DFS for Eastern and Western studies at 1 year (42% vs. 28%) (HR 1.26, 0.85–1.86, *p* = 0.25, I^2^ 4.8%) and 3 years (25% vs. 21%) (HR 0.69, 0.43–1.11, *p* = 0.13, I^2^ 46.2%) were similar, while the 5-year DFS was better in the Eastern studies (25% vs. 10%) (HR 0.29, 0.15–0.54, *p* = 0.0001, I^2^ 77.9%).

The analysis of the funnel plots showed the presence of a slight asymmetry. However, even if the presence of a small study bias cannot be excluded at all, the data suggest that it seems unlikely.

Heterogeneity was significant, so the variability cannot be related only to ethnicity.

#### 3.3.3. Analysis of Prognostic Factors

The results of the meta-analysis of prognostic factors are shown in Table 4 and Table 5.

Nevertheless, we could not include all the studies due to missing data. The analysis demonstrated that the primary cancer factors associated with higher OS were well-moderately differentiated tumors, pT1–2 and pN0–1 adenocarcinoma. Chemotherapy was also a strong prognostic factor as well as R0 resections. Considering the burden of the disease, the presence of solitary metastasis, unilobar and metachronous metastases were all strongly positively associated with OS. On the contrary, older age, sex, size of primary tumor or metastasis, and the presence of lymphatic or venous invasion by the primary tumor were not significantly associated with OS.

The factors associated with a higher DFS were pT1–pT2 primary cancers, the absence of lymphatic invasion by the primary tumors, metachronous liver metastases, solitary metastasis, and the size of liver metastasis <3 cm. However, for the analysis of the prognostic factors related to DFS, less than 5 studies reported the results for each outcome.

The majority of the analyses had mild or low levels of heterogeneity, suggesting that the impact of those prognostic factors on OS and DFS is quite similar despite the well-known differences in the studies.

## 4. Discussion

This systematic review and meta-analysis demonstrate that surgical resection of GCLM, in the absence of peritoneal disease, is a safe procedure, with a 1.6% risk of mortality. It can achieve a 5-year OS of 26%. Those results are in line with the results of recent studies. Long et al., in a meta-analysis [61] of 39 studies, showed a 5-year OS rate of 27%. Similarly, Liao et al. [62], in a comparative analysis of 8 retrospective studies between hepatectomy and chemotherapy only, showed better OS in the surgical group with an odds ratio of 0.17 and 0.15 at 1 and 2 years. Up to now, no other meta-analysis has investigated DFS and prognostic factors related to DFS. We found that liver resection was associated with 5-year DFS of 22%.

Those positive results were particularly highlighted in the Asian studies. Historically, GC survival has always been substantially different in Asian and Western countries. However, results from recent systematic review and meta-analysis are discordant. While Gavriilidis et al. [63] showed no significant difference between Eastern and Western countries in terms of OS, Markar et al. [64] reported a better survival rate for the Eastern studies compared to Western. Furthermore, the results on this topic should be interpreted carefully due to differences between populations in terms of characteristics related to primary cancer, population screening, and dietary habits [65].

In clinical practice, patient selection is crucial in order to achieve acceptable mortality and morbidity. However, in addition to the importance of precision medicine, the role of a careful evaluation of each patient by multidisciplinary teams is improving. In fact, a multidisciplinary approach could increase the proportion of patients’ candidates to curative treatment also in the metastatic setting. In this context, tools to perform a better selection of candidates for hepatectomy, such as the assessment of patients’ performance status, hepatic invasiveness, and the feasibility of obtaining R0 resection, play a central role. Although there are still discordant results in the literature, we believe that radical resection of primary tumor and metastases is vital to achieve good outcomes.

Beom et al. demonstrated in a retrospective study the central role of conversion surgery following chemotherapy for 101 patients with metastatic GC [66]. In the trial, 65 patients (64.4%) had a major response and 11 patients (10.9%) received metastasectomy. Fifty-seven patients (56.4%) had a complete macroscopic resection, with a median survival of 26 months. The importance of R0 resection clearly emerged also in the study of Morgagni et al. [67], in which 11/54 metastatic GC underwent R0 resection. The authors concluded that conversion surgery in metastatic GC could be beneficial only if R0 resection could be achieved. We showed the same results in our analysis, with R1 resection strongly associated with poor OS and DFS. On the other hand, Cheon et al. [68] observed a similar survival rate after R0 or R1 resection, in contrast with our findings. However, the fact that a higher percentage of patients in the Korean series (88% vs. 27.6% in our series) received chemotherapy could be responsible for the differences in the outcomes, due to the positive impact of active treatment on survival. In general, in the case of synchronous disease, achieving an R0 resection both for primary tumor and for the hepatic metastases is recommend. Median survival exceeds 16 months after radical resections and drops to 6 months in case of R1 resections [8]. In our meta-analysis, R0 resection was found strongly associated with a better outcome, in terms of both OS and DFS. In addition to radical resection, other factors have been demonstrated to have a huge impact on prognosis.

In the study of Kinoshita et al. [35], a significant association with poor OS was found in the case of lymphatic and serosal invasion, when the liver metastases were more than 3, the maximum liver metastasis diameter was > 5 cm, or when there was high baseline CEA and CA19.9. Tiberio et al. [8], in an Italian cohort of 105 patients, showed that T-stage, R0 resection, and use of adjuvant chemotherapy were prognostic factors. Montangini et al. [69] showed that T and N staging, lymphovascular invasion of the primary tumor, and the burden of liver disease (i.e., number and diameter of the metastases) were strongly linked with survival. In particular, patients with involvement of the serosa or lymph nodes as well as with lymphovascular invasion had a poor prognosis. Otherwise, ≤3 liver metastases, tumor maximum diameter <5 cm, metachronous presentation, and R0 resections were linked to a better prognosis.

Other authors showed that some clinical and pathologic parameters, such as nodal status and histologic grade of the primary tumor, could impact the prognosis [48]. In this context, the lymph node ratio (number of metastatic lymph nodes on the number of the lymph nodes removed by surgery) is recognized to be an important factor linked to a poor prognosis among patients with GCLM who received combined surgical resection [70,71]. In fact, high lymph node ratio was significantly related to the more advanced pN stage, larger primary tumor dimension, microvascular invasion, and neoadjuvant chemotherapy.

In case of peritoneal involvement from GC, the survival is extremely poor. In those patients, hepatic resection does not add any benefit to survival [72,73]. For this reason, we excluded from the analysis the studies that included patients with peritoneal metastases.

Previous studies [22,56,74] have shown that the presence of multiple or synchronous liver metastases [74] were significant negative prognostic factors. However, in a recent meta-analysis, Cui et al. [75] showed that only the presence of synchronous GCLM was a negative prognostic factor, but they concluded that synchronous or multiple liver metastases should not be considered absolute contraindications for surgery.

Better survival could be obtained in presence of multiple scattered metastases in both lobes if R0 resection can be obtained [8]. In our study, the size of primary tumor and liver metastases were not unfavorable prognostic factors, as well as the presence of lymphatic or venous invasion by the primary tumor. However, we found that the best candidate for aggressive treatment is the patient with a well-moderately differentiated, pT1–2 and pN0–1 primary tumor and solitary, unilobar, and metachronous metastasis. Of course, R0 resection is vital in all cases.

Thus, three main treatment options have been identified for resectable GCLM: chemotherapy, upfront surgery, and preoperative chemotherapy followed by surgery [76]. In Western countries, there is no difference in the treatment strategy in case of synchronous or metachronous GCLM, whereas in Japan, upfront surgery is the preferred treatment option for the synchronous disease. According to international guidelines [5,77], a multimodality approach based on preoperative chemotherapy followed by surgery is considered the best treatment for both synchronous and metachronous resectable GCLM.

Although the REGATTA trial [77] is often cited in support of those guidelines, nowadays many oncologists still do not take the surgical approach into consideration in stage IV GC based on those results. In this phase 3 trial, 175 patients with GC and a single metastatic site confined to liver, peritoneum, or para-aortic lymph nodes were randomized to receive chemotherapy alone or gastrectomy followed by chemotherapy. The study failed to show an improvement in the survival of the experimental arm; additionally, it showed a detrimental effect of gastrectomy in this setting (median OS: 14.3 versus 16.6 months in the chemotherapy arm). However, those results should be interpreted with caution since two main concerns, at least, emerged: first, patients in the REGATTA trial did not receive neoadjuvant chemotherapy; second, the trial was designed to assess only the role of primary tumor site resection (gastrectomy with D1 lymphadenectomy) without any resection of metastatic lesions. That said, it is clear that the surgical treatment in the trial cannot be considered curative but only palliative. Additionally, in the trial, only 9% of the metastatic GC patients had a liver limited disease. Therefore, for all these reasons, the REGATTA trial should not be considered as strong evidence to discharge a multimodality strategy for patients with GCLM.

We believe that for surgeons a new era started since FLOT regimen (5FU, Folinic acid, Oxaliplatin, Docetaxel) was widely approved as the standard of care for advanced GC [78]. The neoadjuvant strategy was widely accepted for >T1N+, and oligometastatic GC has been increasingly recognized as a distinct clinical entity. It is characterized by limited metastatic spread and benefits from a multimodality strategy including chemotherapy and surgery [79]. Despite some limitations due to a non-randomized trial and a relatively small sample size, the phase 2 AIO/FLOT3 [58] showed an increased OS in oligometastatic GC patients treated with neoadjuvant chemotherapy (FLOT schedule) followed surgery. The trial underlined the concept that a more aggressive treatment strategy including preoperative chemotherapy and surgical resection of metastases might have better results in terms of survival in this setting.

Still, open questions are unsolved on which sub-population could really benefit from this strategy. In this regard, the results of the ongoing phase 3 RENAISSANCE/AIO-FLOT5 trial (NCT02578368) [80] are awaited.

In general, our study showed some interesting results that support the new surgical trend of approaching GCLM. However, the importance of neoadjuvant chemotherapy and the number of cycles of adjuvant chemotherapy should be taken into account. In our study, poor data were found for neoadjuvant CT to draw any firm conclusion, although, in our forest plot analysis regarding chemotherapy, including adjuvant and neoadjuvant, HR was 1.49 (*p* = 0.008). If we analyze separately, adjuvant chemotherapy had HR 1.5 (*p* = 0.01) instead of neoadjuvant CT with HR 1.39 (*p* = 0.34), therefore not significant (Figure 7, Appendix A).

Systemic chemotherapy remains the mainstay for the treatment of metastatic GC and progression during chemotherapy is probably the most relevant contraindication for any surgical approach.

In a study focusing on GCLM responder to induction chemotherapy, an R0 resection rate of 100% was obtained in all patients who underwent radical gastric resection plus liver metastasectomy after Docetaxel-Cisplatin-5-FU (DCF) chemotherapy [81]. Kinoshita et al. [82] reported a case series of 18 patients with liver metastases from GC and treated with DCF. In this experience, the majority of patients underwent conversion gastrectomy after chemotherapy (11 patients; 5 patients received also liver metastasectomy), showing an improvement in MST and 3-years OS rate if compared to patients who did not receive surgery (MST: 18.9 vs. 15.6 months; 3-year OS rate: 40.4% vs. 27.5%)

Another retrospective case series including 29 patients with GCLM reported that six patients underwent conversion surgery after DCF chemotherapy (complete response in two patients), two patients received partial hepatectomies with a complete pathological response and two were treated with radiofrequency ablation (RFA) [83]. Additionally, Yamaguchi et al. [84] showed a conversion rate of 21.5% in patients with GCLM liver treated with chemotherapy and subsequent metastasectomy. Same results regarding the good impact of surgery on the survival of those patients were reported in different analyses [66].

Regarding morbidity, most of the studies included in our analysis did not specify if the complications were major or minor. The 30-day morbidity and mortality were 24.7% and 1.6%, respectively, in our study.

Low mortality rate and a limited morbidity rate are not requisites to push patients with GCLM towards a surgical strategy, but certainly these data, in agreement with our results in terms of OS and DFS, reinforce the importance of future randomized prospective studies, needed to validate this strategy and recognize subgroups of patients who can really benefit from it.

Like previous works available in the literature, our study has some limitations. First, all the studies included in the analysis are retrospective and characterized by heterogeneous patient groups. Then, the results may have been affected by selection, institutional and national bias, underpowered sample size, and smaller oncological burden.

## 5. Conclusions

Japanese guidelines have begun to change attitudes towards GCLM, reporting surgical resection as recommended for cases with small number of metastases with no other incurable factor even though with a weak level of evidence [7,85]. Our findings suggest the possibility of expanding surgical indications for patients with GCLM if R0 can be achieved, always in a multidisciplinary setting.

Resection of GCLM is feasible, and a benefit in terms of long-term survival emerged despite the current guidelines. Identification of patient subgroups that would benefit from surgery is mandatory and in a multidisciplinary setting, we should take into consideration the stage of primary cancer, mainly with regard to serosal infiltration and lymph node ratio, and the characteristics of the liver metastases. The presence of solitary or unilobar metastases are positive prognostic factors, whereas size does not matter if GCLM are resectable and R0 could be achieved. Pre- and postoperative chemotherapy plays a key role in the treatment of these patients, even though the role of neoadjuvant CT should be better investigated. The ongoing FLOT 5 trial and a prospective register in the coming years will specify these findings and probably will change the current guidelines. A European registry from which a randomized controlled trial could be developed is necessary in the near future.

## Figures and Tables

**Figure 1 jcm-10-01141-f001:**
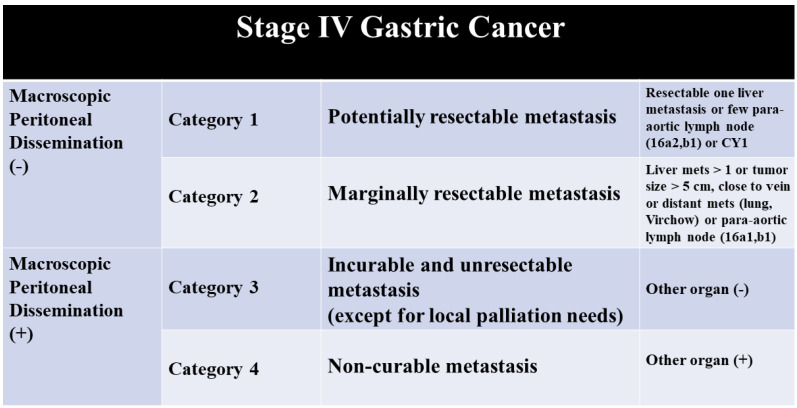
Yoshida categories for stage IV gastric cancer (GC).

**Figure 2 jcm-10-01141-f002:**
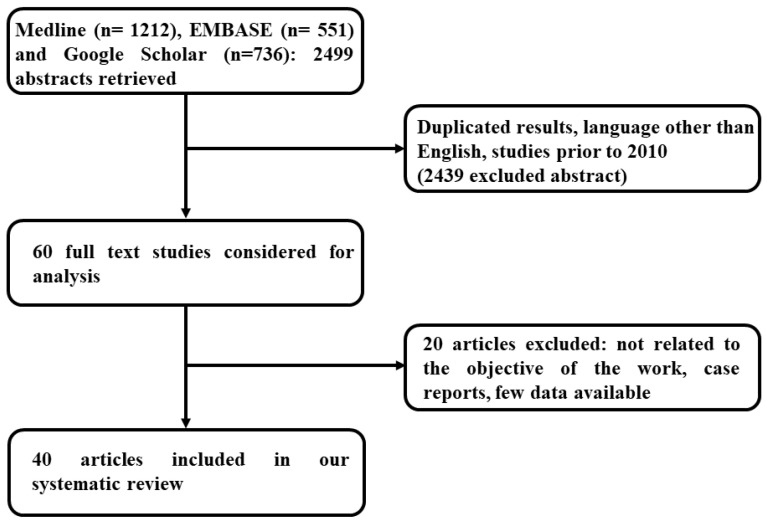
Flow-chart diagram of the study selection.

**Figure 3 jcm-10-01141-f003:**
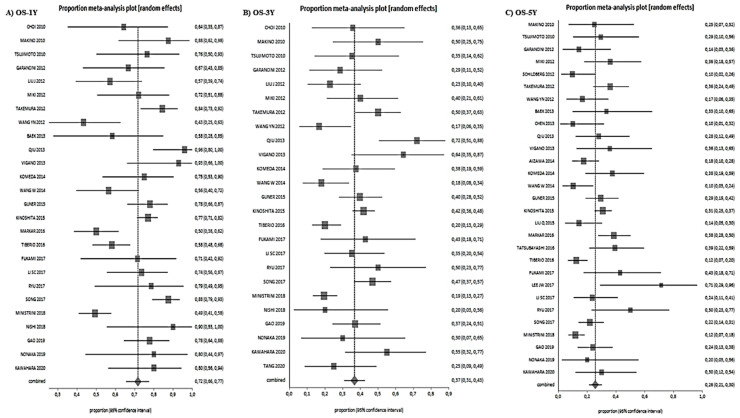
Overall survival results. (**A**): 1-year OS, (**B**): 3-years OS, (**C**): 5-year OS.

**Figure 4 jcm-10-01141-f004:**
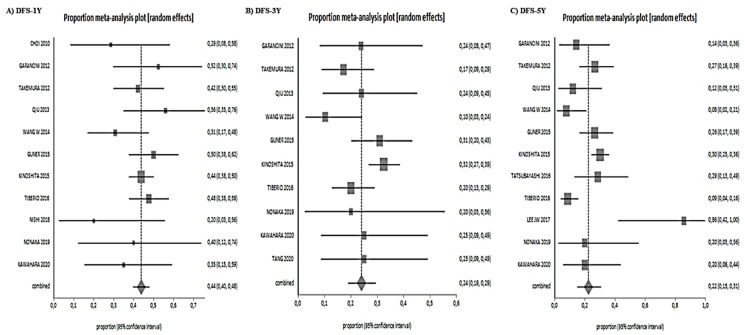
Disease-free survival results. (**A**): 1-year DFS, (**B**): 3-year-DFS, (**C**): 5-year DFS.

**Figure 5 jcm-10-01141-f005:**
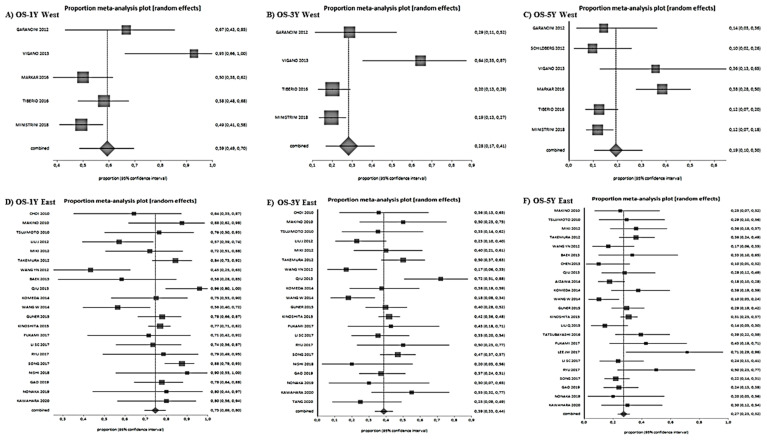
Overall survival according to subgroup analysis (Asian versus Western countries). (**A**): 1-year OS West, (**B**): 3-year OS West, (**C**): 5-year OS West, (**D**):1-year OS East, (**E**): 3-year OS East, (**F**): 5-year OS East.

**Figure 6 jcm-10-01141-f006:**
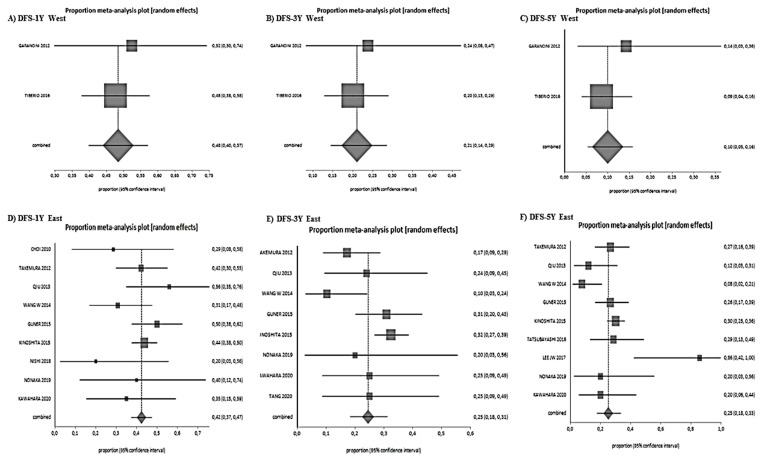
Disease-free survival according to subgroup analysis (Asian versus Western countries). (**A**): 1-year DFS West, (**B**): 3-year DFS West, (**C**): 5-year DFS West, (**D**): 1-year DFS East, (**E**): 3-year DFS East, (**F**): 5-year DFS East.

**Figure 7 jcm-10-01141-f007:**
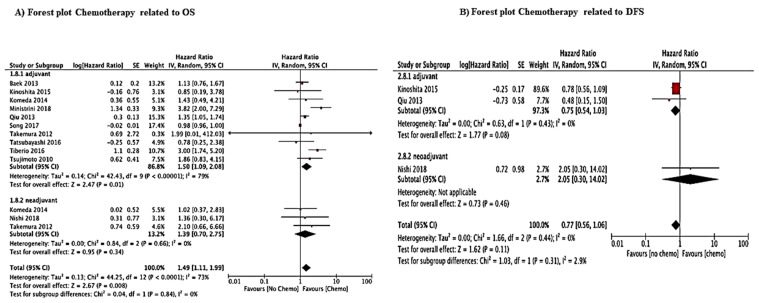
Forest plot showing the impact of chemotherapy on overall survival (OS) and disease-free survival (DFS). (**A**): Forest plot Chemotherapy related to OS (**B**): Forest plot Chemotherapy related to DFS.

**Table 1 jcm-10-01141-t001:** Patient characteristics according to the primary tumor.

Author Year	Country	Study Design	Study Interval	N. Patients	Median Age	Female/Male	pT1–2/pT3–4	pN0–1/pN2–3	Lymphatic Invasion	Venous Invasion	Primary Tumor Median Size (cm)	Histology Well-Moderate/Poor Differentiated
Choi 2010 [57]	Korea	Retro	1986–2007	14	65	NR	NR	NR	NR	NR	NR	NR
Makino 2010 [18]	Japan	Retro	1992–2007	16	NR	3/13	NR/8	NR/13	12	14	NR	8/8
Tsujimoto 2010 [19]	Japan	Retro	1980–2007	17	66	1/16	12/5	12/5	8	9	5.7	NR
Dittmar 2012 [30]	Germany	Retro	1995–2009	10	57	NR	NR	NR	11	NR	NR	NR
Garancini 2012 [41]	Italy	Retro	1998–2007	21	64	7/14	NR/8	19/11	NR	13	NR	8/13
Liu J. 2012 [51]	China	Retro	1995–2010	35	NR	8/29	19/16	12/23	10	NR	NR	0/25
Miki 2012 [52]	Japan	Retro	1995–2009	25	72	2/23	8/17	14/11	NR	NR	NR	NR
Schildberg 2012 [53]	Germany	Retro	1972–2008	31	65	11/20	NR	NR	NR	NR	NR	NR
Takemura 2012 [54]	Japan	Retro	1993–2011	64	65	49/15	NR/49	22/42	NR	NR	NR	42/22
Wang Y.N. 2012 [55]	China	Retro	2003–2008	30	60	3/27	4/26	10/20	NR	NR	3.7	NR
Baek 2013 [56]	Korea	Retro	2003–2010	12	61	1/11	3/9	9/3	7	2	NR	9/1
Chen 2013 [20]	China	Retro	2007–2012	20	54	8/12	6/14	12/8	NR	NR	NR	16/4
Qiu 2013 [21]	China	Retro	1998–2009	25	NR	3/22	17/8	4/21	NR	NR	NR	9/16
Vigano 2013 [22]	Italy	Retro	1997–2008	14	61.5	NR	NR	NR	NR	NR	NR	NR
Aizawa 2014 [23]	Japan	Retro	1997–2010	74	66	18/56	NR	NR	NR	NR	NR	NR
Komeda 2014 [24]	Japan	Retro	2000–2012	24	69.5	3/21	17/7	10/14	NR	NR	NR	NR
Wang W. 2014 [25]	China	Retro	1996–2008	39	64	13/26	8/31	23/16	NR	NR	NR	23/16
Guner 2015 [26]	Korea	Retro	1998–2013	68	61	12/56	17/52	32/36	35	36	NR	NR
Kinoshita 2015 [27]	Japan	Retro	1990–2010	256	64	49/207	74	54/204	105	129	NR	173/NR
Li Z. 2015 [28]	China	Retro	2008–2011	13	NR	NR	NR	NR	NR	NR	NR	NR
Liu Q. 2015 [29]	China	Retro	1990–2009	35	56	13/22	6/29	4/31	20	NR	NR	15/20
Ohkura 2015 [31]	Japan	Retro	1985–2014	13	63	0/13	NR	NR	NR	NR	NR	NR
Shinohara 2015 [32]	Japan	Retro	1995–2010	18	NR	NR	NR	NR	NR	NR	NR	NR
Markar 2016 [33]	UK	Retro	1997–2012	78	65	51/7	NR	NR	NR	NR	NR	NR
Oguro 2016 [34]	Japan	Retro	2002–2012	26	69.5	3/23	8/18	NR/8	NR	NR	NR	18/8
Tatsubayashi 2016 [35]	Japan	Retro	2004–2014	28	72	5/23	8/20	3/25	NR	NR	5.6	22/6
Tiberio 2016 [8]	Italy	Retro	1990–2013	105	68	34/71	38/46	36/40	NR	NR	NR	NR
Fukami 2017 [36]	Japan	Retro	2001–2012	14	66	3/11	2/12	NR/11	NR	NR	NR	11/3
Lee J.W. 2017 [37]	Korea	Retro	2000–2014	7	59.2	2/5	NR	NR	NR	NR	NR	NR
Li J. 2017 [38]	China	Retro	2006–2016	30	NR	NR	NR	NR	NR	NR	NR	NR
Li S.C. 2017 [39]	Taiwan	Retro	1996–2012	34	62	11/23	NR	NR	NR	NR	NR	NR
Ryu 2017 [40]	Japan	Retro	1997–2005	14	NR	NR	NR	NR	NR	NR	NR	NR
Song 2017 [42]	China	Retro	2001–2012	96	63	24/72	47/59	28/68	NR	NR	NR	62/34
Ministrini 2018 [43]	Italy	Retro	1990–2017	144	68	50/94	23/93	48/68	NR	NR	NR	13/22
Nishi 2018 [44]	Japan	Retro	2001–2013	10	71.7	1/9	8/2	NR	NR	NR	NR	NR
Shirasu 2018 [45]	Japan	Retro	2004–2015	9	74	1/8	NR	NR	NR	NR	NR	9/NR
Gao 2019 [50]	China	Retro	1975–2013	54	57	11/43	29/25	18/36	NR	NR	NR	NR
Nonaka 2019 [51]	Japan	Retro	2016	10	68	1/9	3/7	7/3	8	NR	NR	NR
Kawahara 2020 [48]	Japan	Retro	2006–2016	20	73.5	7/13	NR/4	8/12	14	13	NR	14/3
Tang 2020 [49]	China	Retro	2008–2018	20	61	4/16	2/18	10/10	NR	NR	NR	0/12

*Abbreviations: retro: retrospective; NR: not reported.*

**Table 2 jcm-10-01141-t002:** Patient Characteristics according to liver metastases.

Author Year	Synchronous/Metachronous	Solitary/Multiple	Unilobar/Bilobar	Median Size Liver Metastases (mm)	Minor/Major Hepatectomy	R0/R1/R2 Liver Resection Margin	Neoadjuvant/Adjuvant Chemotherapy
Choi 2010 [57]	0/14	9/5	NR	NR	NR	NR	NR
Makino 2010 [18]	9/7	9/7	11/5	NR	14/2	NR	5/9
Tsujimoto 2010 [19]	9/8	13/4	NR	48	NR	NR	0/14
Dittmar 2012 [30]	NR	NR	NR	26	8/2	NR	0/NR
Garancini 2012 [41]	12/9	12/9	16/5	30	17/4	19/2/0	NR
Liu J 2012 [51]	NR	NR	NR	NR	NR	NR	NR
Miki 2012 [52]	16/9	18/7	20/5	20	NR	NR	0/10
Schildberg 2012 [53]	17/14	NR	NR	NR	21/10	23/3/5	2/9
Takemura 2012 [54]	32/32	37/27	NR	NR	50/14	55/9/0	18/26
Wang Y.N. 2012 [55]	30/0	22/8	27/3	31	23/7	NR	0/30
Baek 2013 [56]	3/9	10/1	NR	NR	9/3	11/1/0	NR/6
Chen 2013 [20]	20/0	8/12	11/9	41	6/14	NR	20/20
Qiu 2013 [21]	25/0	19/6	21/4	20	NR	NR	4/14
Vigano 2013 [22]	9/5	9/5	NR	NR	NR	NR	8/0
Aizawa 2014 [23]	74/0	NR	NR	NR	NR	53/0/21	NR
Komeda 2014 [24]	1/23	17/	NR	NR	NR	NR	11/15
Wang W. 2014 [25]	39/0	31/8	34/5	28	NR	NR	0/39
Guner 2015 [26]	26/42	45/23	60/8	27	47/21	NR	0/66
Kinoshita 2015 [27]	106/150	168/88	NR	30	183/73	230/26/0	45/84
Li Z. 2015 [28]	13/0	NR	NR	NR	NR	NR	13/NR
Liu Q. 2015 [29]	35/0	27/8	30/5	NR	29/6	30/5/0	0/35
Ohkura 2015 [31]	9/4	4/9	NR	NR	NR	NR	0/12
Shinohara 2015 [32]	NR	NR	NR	NR	NR	NR	NR
Markar 2016 [33]	78/0	NR	NR	NR	66/12	NR	NR
Oguro 2016 [34]	6/20	16/10	NR	37	NR	NR	NR
Tatsubayashi 2016 [35]	15/13	20/8	NR	24.5	27/1	NR	3/12
Tiberio 2016 [8]	74/31	NR	NR	NR	94/11	89/7/9	0/29
Fukami 2017 [36]	1/13	9/5	NR	28	NR	NR	NR/14
Lee J.W. 2017 [37]	NR	5/2	6/1	NR	NR	NR	0/6
Li J. 2017 [38]	NR	NR	NR	NR	NR	30/0/0	NR
Li S.C. 2017 [39]	0/34	NR	NR	NR	NR	NR	NR
Ryu 2017 [40]	NR	NR	NR	42	7/7	NR	NR
Song 2017 [42]	59/37	42/54	57/29	NR	61/35	91/5/0	0/58
Ministrini 2018 [43]	112/32	NR	NR	NR	132/12	117/14/13	20/32
Nishi 2018 [44]	6/4	6/4	NR	23.5	5/5	NR	2/6
Shirasu 2018 [45]	6/3	0/9	5/4	25	NR	9/0/0	NR/3
Gao 2019 [50]	NR	38/16	NR	NR	NR	NR	0/24
Nonaka 2019 [51]	4/6	7/3	NR	NR	NR	NR	0/0
Kawahara 2020 [48]	11/9	11/9	NR	25	NR	NR	0/20
Tang 2020 [49]	19/1	NR	17/3	29	NR	NR	0/17

*Abbreviations: NR: not reported.*

**Table 3 jcm-10-01141-t003:** Short- and long-term outcomes after hepatectomy.

Author Year	Post-Operative Morbidity (%)	Post-Operative 30-Day Mortality (%)	Overall Survival				Disease-Free Survival			
			1 Year (%)	3 Years (%)	5 Years (%)	MST (Months)	1 Year (%)	3 Years (%)	5 Years (%)	MST (Months)
Choi 2010 [57]	NR	NR	67	38.3	NR	NR	28.5	NR	NR	NR
Makino 2010 [18]	NR	0	82.3	46.4	37.1	31.2	NR	NR	NR	NR
Tsujimoto 2010 [19]	NR	NR	75	37.5	31.5	34	NR	NR	NR	NR
Dittmar 2012 [30]	NR	0	NR	NR	NR	NR	NR	NR	NR	NR
Garancini 2012 [41]	19	0	68	31	19	11	51	25	14	NR
Liu J. 2012 [51]	NR	NR	58.1	21.7	NR	15	NR	NR	NR	NR
Miki 2012 [52]	NR	NR	73.9	42.8	36.7	33.4	NR	NR	NR	5
Schildberg 2012 [53]	29	6	NR	NR	13	NR	NR	NR	NR	NR
Takemura 2012 [54]	27	0	84	50	37	34	42	27	27	9
Wang Y.N. 2012 [55]	13	0	43.3	16.7	16.7	11	NR	NR	NR	NR
Baek 2013 [56]	NR	0	65	NR	39	31	NR	NR	NR	NR
Chen 2013 [20]	NR	0	NR	NR	15	22.3	NR	NR	NR	NR
Qiu 2013 [21]	NR	0	96	70.4	29.4	38	56	22.3	11.1	18
Vigano 2013 [22]	40	0	95	63.2	33.2	52.3	NR	NR	NR	NR
Aizawa 2014 [23]	NR	0	NR	NR	17	13	NR	NR	NR	NR
Komeda 2014 [24]	NR	0	78.3	40.1	40.1	22.3	NR	NR	NR	NR
Wang W 2014 [25]	8	0	56.4	17.9	10.3	14	30.8	10.3	7.7	8
Guner 2015 [26]	28	1	79.1	40.6	30	24	49.3	30.4	26	NR
Kinoshita 2015 [27]	11	2	77.3	41.9	31.1	31.1	43.6	32.4	30.1	9.4
Li Z. 2015 [28]	NR	NR	NR	NR	NR	16.3	NR	NR	NR	NR
Liu Q. 2015 [29]	6	0	NR	NR	14.3	33	NR	NR	NR	NR
Ohkura 2015 [31]	NR	0	NR	NR	NR	NR	NR	NR	NR	NR
Shinohara 2015 [32]	NR	0	NR	NR	NR	NR	NR	NR	NR	NR
Markar 2016 [33]	NR	10	64.1	NR	38.5	NR	NR	NR	NR	NR
Oguro 2016 [34]	NR	NR	NR	NR	13.9	20.1	NR	NR	NR	16.8
Tatsubayashi 2016 [35]	4	0	NR	NR	32	49	NR	NR	29	NR
Tiberio 2016 [8]	13	1	58.2	20.3	13.1	14.6	48	20.2	8.6	10
Fukami 2017 [36]	21	0	71.4	42.9	42.9	27.9	NR	NR	NR	NR
Lee J.W. 2017 [37]	29	NR	NR	NR	68.6	67.5	NR	NR	80	74.1
Li J. 2017 [38]	NR	0	-	-	-	-	-	-	-	-
Li S.C. 2017 [39]	NR	NR	73.5	36.9	24.5	26.16	NR	NR	NR	NR
Ryu 2017 [40]	NR	0	84.6	51.3	51.3	NR	NR	NR	NR	NR
Song 2017 [42]	55	0	87.5	47.6	21.7	34	NR	NR	NR	NR
Ministrini 2018 [43]	22	2	49.5	19.4	11.6	12	NR	NR	NR	NR
Nishi 2018 [44]	10	0	88.9	17.8	NR	21.5	20	NR	NR	4.7
SHIRASU 2018 [45]	44	0	NR	NR	NR	24.8	NR	NR	NR	7.9
Gao 2019 [50]	NR	NR	77.8	37	25.9	29.3	NR	NR	NR	NR
Nonaka 2019 [51]	NR	NR	78	33.3	22.2	30	44.4	22.2	22.2	NR
Kawahara 2020 [48]	0	0	80	55.5	31.7	42	35	24	18	10.5
Tang 2020 [49]	25	15	NR	23.5	NR	20	NR	23.5	NR	NR

*Abbreviations: MST: median survival time; NR: not reported.*

**Table 4 jcm-10-01141-t004:** Analysis of prognostic factors related to overall survival.

Prognostic Factor	N. of Studies	Participants	HR (95% CI)	*p*	I^2^
Male sex	16	788	0.86 (0.68–1.09)	0.21	0%
Age >65	19	896	0.86 (0.49–1.51)	0.60	94%
Synchronous liver metastases	14	730	1.62 (1.17–2.25)	0.004	62%
Multiple liver metastases	17	788	1.66 (1.44–1.91)	<0.00001	4%
Bilobar liver metastases	9	495	1.96 (1.34–2.87)	0.0005	69%
>3 cm liver metastases	19	803	2.39 (1.14–5.04)	0.02	98%
R + liver resection margin	6	400	4.15 (2.37–7.26)	<0.00001	33%
Chemotherapy before/after liver resection	11	781	1.49 (1.11–1.99)	0.008	73%
Primary tumor Size >5 cm	7	179	1.50 (0.99–2.26)	0.06	14%
pT3–4	21	1084	1.77 (1.31–2.41)	0.0002	51%
pN2–3	16	750	1.54 (1.28–1.85)	<0.00001	11%
Lymphatic invasion present	9	467	1.28 (0.96–1.70)	0.09	72%
Venous invasion present	7	364	1.23 (0.93–1.62)	0.15	0%
Primary tumor poorly differentiated	17	796	1.34 (1.10–1.63)	0.004	14%

**Table 5 jcm-10-01141-t005:** Analysis of prognostic factors related to disease-free survival.

Prognostic Factor	N. of Studies	Participants	HR (95% CI)	*p*	I^2^
Male sex	3	291	0.94 (0.65–1.36)	0.76	0%
Age >65	4	301	0.96 (0.70–1.31)	0.80	39%
Synchronous liver metastases	4	302	1.50 (1.21–1.86)	0.0002	0%
Multiple liver metastases	4	301	2.34 (1.67–3.29)	<0.00001	0%
Bilobar liver metastases	1	25	3.39 (1.09–10.56)	0.04	-
>3 cm liver metastases	4	301	1.51 (1.10–2.07)	0.01	0%
Chemotherapy before/after liver resection	3	291	0.77 (0.56–1.06)	0.11	0%
Primary tumor Size >5 cm	1	10	3.22 (0.71–14.57)	0.13	-
pT3–4	4	301	1.43 (1.06–1.94)	0.02	0%
pN2–3	3	292	1.35 (0.93–1.97)	0.11	35%
Lymphatic invasion present	3	291	1.46 (1.02–2.08)	0.04	43%
Venous invasion present	2	266	1.25 (0.92–1.70)	0.16	0%
Primary tumor poorly differentiated	4	317	1.27 (0.80–2.01)	0.31	46%

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
