# Peer review of "Efficacy of Surgery for the Treatment of Gastric Cancer Liver Metastases: A Systematic Review of the Literature and Meta-Analysis of Prognostic Factors"

_jcm, 2021, doi:10.3390/jcm10051141_

Round 1

Reviewer 1 Report

The Review article authored by Dr Marte et al aims to evaluate the efficacy of hepatectomy in the management of patients with gastric cancer liver metastasis (GCLM), and to delineate favorable prognostic factor in the definition of the subgroup of patients who will benefit from surgery. This review is a meta-analysis of 40 retrospective studies performed from 2010 to 2020 that encompasses 1573 patients who underwent hepatic resection for GCLM. The authors clearly explained the selection of the studies, and detailed the collection and analysis of the data. This extensive literature analysis of the literature indicates that hepatectomy for GCLM is feasible and could provide benefit in terms of long-term patient survival. Furthermore, it suggests that some pathological and clinical parameters, e.g. tumor size and differentiation, lymph node involvement, resection of solitary or unilobar metastasis might constitute positive prognostic factors for patients subgroup who will benefit from a surgical treatment. Considering the poor outcome of metastatic gastric cancers and the actual guidelines that recommend systemic chemotherapy as single modality treatment, this study underlies the requirement of multidisciplinary setting in the management of these cancers and the revaluation of actual approaches based on the results of ongoing clinical trials. This manuscript is well written and illustrated. This review would make therefore a suitable contribution for JCM.

Minor

The title of the manuscript “Role of  …” is a little bit ambiguous. “Efficacy of “ or “Benefit of” would be more accurate.

According to other papers published in JCM, the first name of the Authors should precede the family name. This is the opposite in the 1st page of the manuscript.

Lines 68,69 According to the classification of Yoshida et al, both categories 1 and 2 of GCLM belong to the subgroup gastric cancer liver metastasis without peritoneal carcinomatosis (not only category 2)

The Authors should provide the acronym of some abbreviations, e.g. MST, FLOT, RFA

The text in Figures 3, 4, 5, 6 and 7 is difficult to read. Please, increase the size of the police.

Supplementary Materials contain 18 plots instead of mentioned 25 figures (lines 483-496). Please, provide the numbers and titles to the Supplementary Figures.

The Authors need to read carefully the manuscript to correct some misprints:

Line 34: change “meta-analysis is evaluate’ to ‘meta-analysis is to evaluate”

Line 116: change “has changed significantly the last decade” to “has changed significantly during the last decade”

Line 144: change “iterature search” to “Literature search”

Line 181: change “I2 statisticto” to “I2 statistict”

Table 2, 5th column: change “Median size liver metastases (cm)” to “Median size liver metastases (mm)”

Line 235: change “tumor >3 cm” to “tumor <3 cm”

Author Response

Reviewer 1: English language and style are fine/minor spell check required=> the paper has been revised by a english mother tongue colleague.

  • The title of the manuscript “Role of …” is a little bit ambiguous. “Efficacy of “ or “Benefit of” would be more accurate.=> in line 3 title has been changed as suggested in Efficacy of surgery....
  • According to other papers published in JCM, the first name of the Authors should precede the family name. This is the opposite in the 1st page of the manuscript.=> Line 7-8 corrected
  • Lines 68,69 According to the classification of Yoshida et al, both categories 1 and 2 of GCLM belong to the subgroup gastric cancer liver metastasis without peritoneal carcinomatosis (not only category 2)=> line 71 corrected in categories 1 and 2.
  • The Authors should provide the acronym of some abbreviations, e.g. MST, FLOT, RFA=> corrected in line 164-165 (MST), line 235-236 (RFA etc...) 
  • The text in Figures 3, 4, 5, 6 and 7 is difficult to read. Please, increase the size of the police.=> all figures has been increased in term of size of police (minimum 300 dpi).
  • Supplementary Materials contain 18 plots instead of mentioned 25 figures (lines 483-496). Please, provide the numbers and titles to the Supplementary Figures.=> remaining plots have been uploaded (corrected in 24 figures)

The Authors need to read carefully the manuscript to correct some misprints:

Line 34: change “meta-analysis is evaluate’ to ‘meta-analysis is to evaluate”=> corrected in is to assess

Line 114: change “has changed significantly the last decade” to “has changed significantly during the last decade”=> corrected in over the last decade

Line 142: change “iterature search” to “Literature search”=> corrected

Line 174: change “I2 statisticto” to “I2 statistict”=> corrected

Table 2, 5th column: change “Median size liver metastases (cm)” to “Median size liver metastases (mm)”=> corrected

Line 225: change “tumor >3 cm” to “tumor <3 cm”=> corrected

Finally thanks for your improving our paper, introduction has been improved as you request and results better presented. 

Sincerely, Gianpaolo Marte.

Reviewer 2 Report

Authors clearly showed systematic review of GCLM. The prognosis was worse than CRLM. However, there were some populations who achieved long-term survival. Authors clearly showed the prognostic factor after liver resection for GCLM. Therefore, this article might be beneficial for liver surgeon, oncologist, and the patients with GCLM. 

Author Response

Thanks a lot for your comment on our paper, we have improved with an extensive revision of english language and some other. I hope you will accept it for publishing. 

Sincerely Gianpaolo.

Reviewer 3 Report

Dear Dr. Marte,

I congratulate you for this excellent work. 

Few minor comments:

Gastric cancer liver metastasis (GCLM) without peritoneal carcinomatosis belong to category 2.  However the  the Yoshida classification  category 1 also include  liver met. Please  explain this more in depth why your selection includes only category 2

I believe  it is important to outline in introduction and discussion what  impact  stage IV  diagnosis  has  and why surgery needs to be  more focused into the therapy regimen nowadays. a few lines on the ignorance of  some oncologist and surgical oncologist is necessary  to  emphasize  that even nowadays many centres do not consider these options.

If i got it right you have excluded FLOT trials from  inclusion . this is  from my point of view  not correct as there was a subgroup of patients with only liver mets in that trial and also  FLOT is considered primrary approach in  limited  stage IV in germany and other few european countries.

Author Response

Dear reviewer, 

an extensive revision of english language has been done by an English colleague. 

Point by point answer:

: English language and style are fine/minor spell check required

  • Gastric cancer liver metastasis (GCLM) without peritoneal carcinomatosis belong to category 2. However the  the Yoshida classification  category 1 also include  liver met. Please  explain this more in depth why your selection includes only category 2=> line 71, category 1 and 2 , corrected

  • I believe it is important to outline in introduction and discussion what  impact  stage IV  diagnosis  has  and why surgery needs to be  more focused into the therapy regimen nowadays. a few lines on the ignorance of  some oncologist and surgical oncologist is necessary  to  emphasize  that even nowadays many centres do not consider these options.=> line 59-63, introduction as been improved and line 381-382 added ("ignorance of some oncologist")

  • If i got it right you have excluded FLOT trials from inclusion . this is  from my point of view  not correct as there was a subgroup of patients with only liver mets in that trial and also  FLOT is considered primrary approach in  limited  stage IV in germany and other few european country=> we have explained why FLOT was excluded , because it was not possible to extract data for the subgroup with liver mets only.

Thanks a lot for your comments that we are sure are improving our paper, I hope you will accept it for publishing.

Sincerely Gianpaolo.

Reviewer 4 Report

The present study showed that hepatectomy for unresectable gastric cancer patients is feasible and could provide benefit in long-term survival.

I think that there are not any problem about methods and investigations in this manuscript.

The discussion is well considered.

Author Response

Dear Reviewer,

thank a lot for your comment on our article, an extensive revision of english language and style ha been made. 

I hope you will accept it for publishing. 

Sincerely, Gianpaolo. 

Round 2

Reviewer 3 Report

-